# Prevalence and associated risk factors of general and abdominal obesity in rural and urban women in Bangladesh

**Farjana Islam** *, **Rahanuma Raihanu Kathak, Abu Hasan Sumon, Noyan Hossain Molla**

Department of Biochemistry and Molecular Biology, Shahjalal University of Science and Technology, Sylhet, Bangladesh

* farjanaislam308@gmail.com, farjana-bmb@sust.edu

## Abstract

### Background

Obesity is a major public health concern worldwide including Bangladesh. This study aimed to assess the prevalence and associated risk factors of general and abdominal obesity in rural and urban women in Bangladesh.

### Methods

A total of 450 adult women aged $\geq$ 18 years were recruited from rural (n = 210) and urban (n = 240) areas of four administrative regions (Chattagram, Dhaka and Rajshahi and Sylhet) of Bangladesh. Both socio-demographic and anthropometric data were recorded in this study. WHO proposed cut-off values were used for the Asian population for defining general and abdominal obesity. Multinomial logistic regression analysis was applied to evaluate the risk factors of general and abdominal obesity for Bangladeshi women.

### Results

Overall, the prevalence of general and abdominal obesity was 28% and 49%, respectively. Urban women had a significantly higher prevalence of both general and abdominal obesity (30.9% and 58.6%, respectively) than in the rural women (26.6% and 38.1%, respectively) (p<0.05 and p<0.01, respectively). As region comparison, the prevalence of general obesity was higher in the Dhaka region (39.3%) compared to the Chattragram (23.3%), Rajshahi (23.9%) and Sylhet (3.5%) regions. On the other hand, abdominal obesity was more frequent among participants in Sylhet (72.4%) and Dhaka regions (61.5%), compared to the Chattagram (27.4%) and Rajshahi (37.3%) regions. A wide variation has been observed on the prevalence of general and abdominal obesity in the different age groups of the four regions. In regression analysis, a high socioeconomic status (ref: low socioeconomic level), low education level (ref: higher education), low physical activity (ref: adequate physical activity) and middle age (ref: $\geq$ 30 years of age) were significant risk factors for general and abdominal obesity.

**Data Availability Statement:** All relevant data are within the paper.

**Funding:** The author(s) received no specific funding for this work.

**Competing interests:** The authors have declared that no competing interests exist.

## Conclusions

The prevalence of general and abdominal obesity was higher among participants living in urban areas. Physical inactivity, middle age, high socioeconomic status and low education level were associated with the increased prevalence of general and abdominal obesity. Such a high prevalence of general and abdominal obesity is a health concern for Bangladeshi women; therefore, public awareness and effective health intervention strategies are needed to address these conditions.

## Introduction

Obesity is one of the major public health concerns due to its upward trend in both developing and developed countries [1,2]. According to the World Health Organization (WHO), about 39% of adults aged $\geq$18 years were overweight and 13% of adults were obese in the world in 2016 [3]. Based on the WHO report, 1 in 3 of the world's adult population is overweight and 1 in 10 is obese. Undernutrition is more frequent in developing countries, however, over the past two decades, overweight and obesity are increasing rapidly in low to middle-income countries alongside undernutrition [4]. In an epidemiological survey, body mass index (BMI) is used as an indicator of general obesity and waist circumference (WC) and waist-hip ratio (WHR) are used as an indicator of abdominal obesity. In 2010 in Bangladesh, WHO estimated the prevalence of obesity/overweight (BMI $\geq$25 kg/m$^2$) aged >15 years was 8.4% [5]. In another study, age-standardized and BMI based prevalence of obesity was 26.2% and WC based prevalence of abdominal obesity was 39.8% in rural Bangladeshi adults aged 20 years and over [6]. An increasing trend of obesity has been reported in females of reproductive age in South Asia between 1996 and 2006 [7]. In many studies, measures of abdominal obesity have been reported as the best parameter that correlates with cardiovascular disease and metabolic disorders. While other studies did not find sufficient evidence that the measure of abdominal obesity is superior to BMI in predicting the risk of cardiovascular disease [8,9]. In another study, both general and abdominal obesity were associated with several metabolic abnormalities including hypertension, type 2 diabetes, cardiovascular disorders and metabolic syndrome [10].

In developing countries, the consequence of obesity not only affects health but also creates a burden on individual and national healthcare budgets [6]. Bangladesh is a low-income and agro-based country where a major percentage of the national population (72%) lives in rural areas [6]. Bangladesh's population is projected to increase from 164 million in mid-2020 to 192 million in 2050 [11]. Over the past two decades, Bangladesh has experienced a rapid epidemiological transition from communicable diseases to non-communicable diseases [5,12]. Rapid urbanization and industrialization also contribute to the increased prevalence of obesity in the Bangladeshi population. Little is known on the prevalence of general and abdominal obesity in rural and urban women in Bangladesh. In the present study, we aimed to estimate the prevalence of general and abdominal obesity and their associated risk factors in adult women living in rural and urban areas in Bangladesh.

## Methods

### Study area and study participants

This study was a cross-sectional design conducted between October 2016 and June 2017 at the Department of Biochemistry and Molecular Biology of Shahjalal University of Science and

Technology, Bangladesh. In the present study, participants were enrolled from rural and urban communities of the administrative regions of Chattagram, Dhaka (capital city), Rajshahi and Sylhet of Bangladesh. The country has eight administrative regions; of these, the included four regions were the largest administrative regions in Bangladesh. A total of 450 participants (210 rural and 240 urban) were selected from the rural and urban areas of these regions. The female participants aged above 18 years who were residents of these areas and consented to participate were included in the study. Systematic sampling was done and every 10th house-hold was selected for participation. From each house only one available family member was selected for participation. During household selection we considered an equal probability as the selected households could provide a statistically reliable estimate of key demographic and health-related variables. The inclusion criteria were female gender, age ≥18 years, free from severe chronic illness and willingness to participate. Exclusion criteria included the lactating mother, pregnant women and individuals who had cardiac disease, renal disease, liver disease, malignant disease and any severe infection at the time of data collection. We also excluded participants with missing socio-demographic data. All participants were informed about the study aims and written consent obtained from them before inclusion in the study. This study was conducted by following relevant guidelines and the study was reviewed and approved by the Internal Ethics Committee at the Department of Biochemistry and Molecular Biology of the university.

## Data collection and anthropometric measurements

The socio-demographic data was collected by face-to-face interviewing the participants at their homes. Survey procedure included completing a short questionnaire on socio-demographic, food habits, smoking status, physical activity and anthropometric measurements described elsewhere [13–18]. Briefly, anthropometric measurements such as weight, height, waist circumference (WC) and hip circumference (HC) were taken with the subjects wearing light clothes and no shoes. Weight was measured to the nearest 0.1 kg by modern electronic digital LCD weighing machines (Beurer 700, Germany) placed on a flat surface. The weight measuring scales were calibrated regularly against a standard (20 kg). Height was measured to the nearest 0.1 cm while the study subjects stood in the erect posture. Body mass index (BMI) was calculated as the weight in kg divided by height in meter squared ($m^2$). WC was measured by placing the tape horizontally midway between the lowest border of the ribs and iliac crest on the mid-axiliary line. HC was measured at a level parallel to the floor, at the largest circumference of the buttocks to the nearest 0.5 cm. The waist-to-hip ratio (WHR) was then calculated by dividing WC by HC. Data collection and anthropometric measurements were done by trained personnel and the accuracy of the data confirmed by repeated measurements in the presence of investigators.

## Definitions

Based on definition for obesity for Asian population recommended by the WHO, we categorized BMI into four groups: underweight ($< 18.5$ kg/$m^2$), normal weight (18.5–23.0 kg/$m^2$), overweight (23.0–27.5 kg/$m^2$, and obese ($\geq 27.5$ kg/$m^2$) [19]. Central or abdominal obesity was defined as a WC $\geq 90$ cm for males and $\geq 80$ cm for females and WHR $\geq 0.90$ and $\geq 0.80$ for males and females, respectively [20]. Education level graded as illiterate: unable to read and write; having primary and secondary and higher secondary and above. Socioeconomic status was categorized as low ($<10000$ Bangladeshi Taka, BDT) (1 USD = 85 BDT), medium (10000–20000 BDT) and high ($>20000$ BDT) based on their monthly income. Physical activity was classified as low (little housework and comfortable office jobs), medium (swimming, general

walking and cleaning the household goods) and adequate/high (lifting, carrying, jogging and/ or sports). Smoking status was defined as a current smoker and never smoker.

## Statistical analysis

Results are presented as mean ± SD. Differences in the socio-demographic characteristics in the WC and BMI groups were analyzed by independent sample t-test and one-way ANOVA, respectively. A Chi-square test was applied to differentiate the proportions of the categorical variables. Multinomial logistic regression analysis was done to assess the relationship of covariates with general and abdominal obesity. In regression analysis, general obesity and abdominal obesity (yes) were considered as the dependent variable and socio-demographic variables were the independent variable. Socio-demographic factors and physical activity were included in the regression analysis to identify variables independently associated with general and abdominal obesity. Statistical analysis of data was performed using the software IBM SPSS Statistics version 23. A p-value of less than 0.05 was considered statistically significant.

## Results

The socio-demographic characteristics of the participants are presented in Table 1. Among the participants, 210 were rural and 240 were urban participants. The mean age (± SD) of the participants was 38.7 ± 13.6 years. The mean BMI (kg/m$^2$) was comparatively high in participants from Dhaka region (26.3 ± 3.9) compared to Chattragram (25.2 ± 3.8), Rajshahi (23.8 ± 5.1) and Sylhet (21.9 ± 4.4) regions. Overall, a notable portion of the participants was illiterate (16.9%) or having elementary education (23.8%). A major percentage of the participants were housewives (45.3%) and this is usual as most of the women living in rural areas are involved in housework other than jobs. Regarding the socioeconomic status, there were about 30%, 55% and 14% of the participants who had a low, middle and high socioeconomic status. Among the participants, only 4.4% of subjects were physically more active and 32.7% of participants were physically inactive or had very low activity.

The prevalence of general and abdominal obesity is shown in Table 2. The overall prevalence of general and abdominal obesity was 28% and 49%, respectively. As a residence comparison, urban participants had a significantly higher prevalence of both general and abdominal obesity (30.9% and 58.6%, respectively) compared to that found among rural participants (26.6% and 38.1%, respectively) (p<0.05 and p<0.01, respectively). As a region comparison, participants in the Dhaka region had a high prevalence of general obesity (39.3%) and participants in the Sylhet region had a high prevalence of abdominal obesity.

Age-specific prevalence of general and abdominal obesity is shown in Fig 1. A wide variation has been observed on the prevalence of general and abdominal obesity in different age groups of four regions. Individual lifestyle, food habits and the climate might be the considerable factors associated with the observed variations in these regions.

Table 3 represents the relationship of general and abdominal obesity with socio-demographic variables. The middle age, medium and high socioeconomic status, low education level and low physical activity were the significant indicators for general and abdominal obesity in regression analysis. Bangladeshi women are not usually habituated with smoking; therefore, this variable was not included in the regression analysis.

## Discussion

This study was conducted to explore the prevalence of general and abdominal obesity and their associated socio-demographic and lifestyle determinants in rural and urban women of four divisional regions in Bangladesh. In this study, we have applied the WHO proposed cut-

**Table 1. Demographic characteristics of the participants.**

| Variables | Total (n = 450) | Chattagram (n = 115) | Dhaka (n = 164) | Rajshahi (n = 105) | Sylhet (n = 66) |
|---|---|---|---|---|---|
| N | | | | | |
| *Rural* | 210 | 54 | 54 | 58 | 44 |
| *Urban* | 240 | 61 | 110 | 47 | 22 |
| Age (yrs) | 38.7 ± 13.6 | 41.1 ± 11.7 | 34.3 ± 11.2 | 44.1 ± 15.3 | 39.1 ± 17.9 |
| Height (cm) | 153.9 ± 6.1 | 158.3 ± 5.7 | 152.8 ± 5.8 | 152.6 ± 4.7 | 151.1 ± 5.9 |
| Weight (cm) | 59.3 ± 11.1 | 62.7 ± 6.5 | 61.5 ± 10.7 | 55.6 ± 12.3 | 50.4 ± 11.9 |
| BMI (kg/m$^2$) | 24.9 ± 4.4 | 25.2 ± 3.8 | 26.3 ± 3.9 | 23.8 ± 5.1 | 21.9 ± 4.4 |
| WC (cm) | 78.4 ± 14.0 | 72.5 ± 7.6 | 82.4 ± 3.4 | 75.9 ± 17.8 | 82.9 ± 12.8 |
| HC (cm | 91.4 ± 9.5 | 84.7 ± 6.3 | 95.3 ± 9.6 | 91.7 ± 9.8 | 91.8 ± 5.6 |
| WHR | 0.86 ± 0.09 | 0.86 ± 0.08 | 0.87 ± 0.07 | 0.85 ± 0.07 | 0.86 ± 0.18 |
| Education % | | | | | |
| Illiterate | 16.9 | 13.6 | 25.8 | 11.7 | 4.0 |
| Primary | 23.8 | 33.3 | 23.7 | 18.3 | 12.0 |
| High School | 20.2 | 16.7 | 21.7 | 25 | 12.0 |
| Higher Secondary | 8.1 | 9.0 | 5.2 | 8.3 | 16.0 |
| Above Higher Secondary | 31.0 | 27.3 | 23.7 | 36.7 | 56.0 |
| Occupation % | | | | | |
| Housewives | 45.3 | 56.4 | 59.2 | 29.4 | 4.3 |
| Job | 38.2 | 24.2 | 30.3 | 54.9 | 65.2 |
| Others | 16.5 | 19.4 | 10.5 | 15.7 | 30.4 |
| Socio-economic status % | | | | | |
| Low | 30.2 | 29.3 | 31.8 | 40 | 0.0 |
| Middle | 55.4 | 60.3 | 53.4 | 41.8 | 85.7 |
| High | 14.4 | 10.4 | 14.8 | 18.2 | 14.3 |
| Physical activity % | | | | | |
| Low | 32.7 | 0.0 | 49.4 | 90.0 | 15.0 |
| Moderate | 62.9 | 86.4 | 50.6 | 10.0 | 85.0 |
| Adequate/high | 4.4 | 13.6 | 0.0 | 0.0 | 0.0 |

Data are presented as mean ± SD. Categorical values are presented as percentage.

**Table 2. Prevalence of general and abdominal obesity among the participants.**

| Variables | Overall (n = 450) | | | Chattagram (n = 115) | | | Dhaka (n = 164) | | | Rajshahi (n = 105) | | | Sylhet (n = 66) | | |
|---|---|---|---|---|---|---|---|---|---|---|---|---|---|---|---|
| | Rural | Urban | Total | Rural | Urban | Total | Rural | Urban | Total | Rural | Urban | Total | Rural | Urban | Total |
| BMI (kg/m$^2$) | | | | | | | | | | | | | | | |
| Underweight | 11.6 | 2.6 | 7.0 | 0.0 | 2.5 | 1.4 | 0.0 | 2.6 | 1.7 | 27.0 | 3.3 | 16.4 | 25.0 | 0.0 | 20.7 |
| Normal | 26.6 | 24.3 | 24.5 | 11.8 | 38.5 | 26.0 | 25.6 | 15.4 | 18.8 | 29.7 | 30.0 | 29.9 | 33.3 | 20.0 | 30.0 |
| Overweight | 38.8 | 42.1 | 40.5 | 38.2 | 59.0 | 39.3 | 38.5 | 41.0 | 40.2 | 37.8 | 20.0 | 29.0 | 41.7 | 60.0 | 44.8 |
| Obesity | 26.6 | 30.9[a] | 28.0 | 50.0 | 0.0 | 23.3 | 35.9 | 41.0 | 39.3 | 5.4 | 46.7 | 23.9 | 0.0 | 20.0 | 3.5 |
| WC (cm) | | | | | | | | | | | | | | | |
| Normal | 61.9 | 41.4 | 51.0 | 94.1 | 53.8 | 72.6 | 41.0 | 35.9 | 38.5 | 73.0 | 50 | 62.7 | 33.3 | 0.0 | 27.6 |
| Abdominal obesity | 38.1 | 58.6[b] | 49.0 | 5.9 | 46.2 | 27.4 | 58.9 | 62.8 | 61.5 | 27.0 | 50 | 37.3 | 66.7 | 100.0 | 72.4 |

[a]P<0.05 when prevalence of general obesity in urban area is compared to rural area.

[b]P<0.01 when prevalence of abdominal obesity in urban area is compared to rural area. A Chi-square test was applied to differentiate the proportions of the categorical variables.

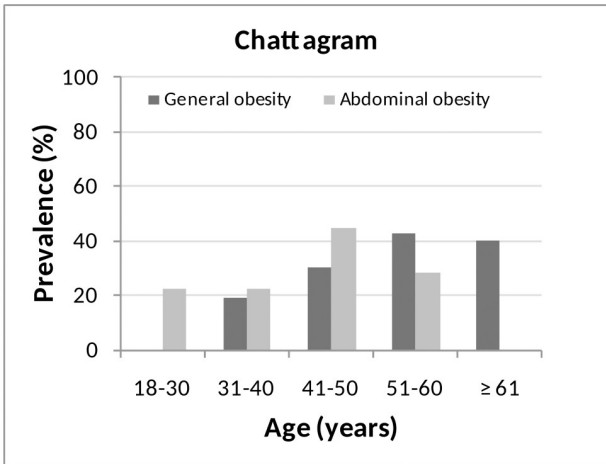 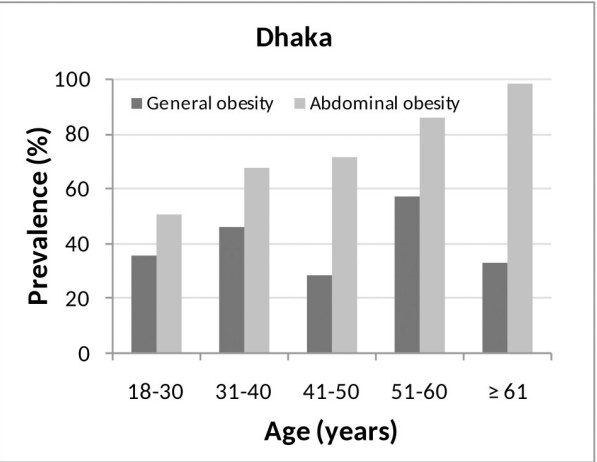

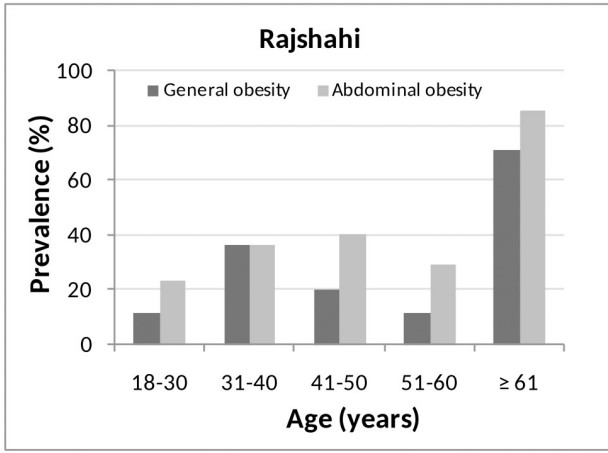 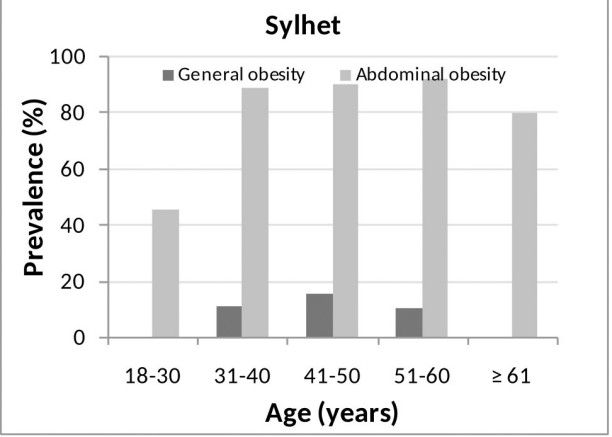

**Fig 1. Prevalence of general and abdominal obesity by age.**

off values for the Asian population for defining general and abdominal obesity. According to WHO, a BMI of 18.5 to 23 kg/m$^2$ is considered healthy for the Asian population [19]. Similarly, low-cut off values have also been recommended by the WHO for the South Asian population [20]. Based on WHO guidelines, abdominal or central obesity is defined as a WC $\geq$ 90 cm for males and $\geq$ 80 cm for females and WHR $\geq$ 0.90 and $\geq$ 0.80 for males and females, respectively [20].

In the present study, the overall prevalence of general obesity was 28%, and this prevalence was higher in urban areas (30.9%) than in rural areas (26.6%). As a region comparison, participants in the Dhaka region had a high prevalence of general obesity (39.3%), whereas, participants in the Sylhet region had a high prevalence of abdominal obesity. Individual lifestyle, food habits and the climate might be the considerable factors associated with these variations in the regions. The prevalence of general obesity documented in the current study is comparatively higher than that reported in the previous studies that applied various BMI cut-off values [6,7]. In a recent study, the prevalence of general obesity was found as 26.8% in rural Bangladeshi women [6] which is comparatively lower than the prevalence found in the present study. The prevalence of general obesity in the present study is also higher than that reported in adults living in the rural areas of India and China [21,22]. In another recent survey, the

**Table 3. Association between general and central obesity and socio-demographic factors in the surveyed population.**

| Variables | General obesity | | Central Obesity | |
|---|---|---|---|---|
| | OR (95% Cl) | P-value | OR (95% Cl) | P-value |
| Age | | | | |
| 18–30 | Ref | | Ref | |
| 31–40 | 1.66 (1.26–2.14) | <0.001 | 1.71 (1.28–2.16) | <0.001 |
| 41–50 | 1.47 (1.12–1.84) | <0.001 | 1.58 (1.24–1.94) | <0.001 |
| 51–60 | 1.42 (1.10–1.78) | <0.01 | 1.48 (1.16–1.84) | <0.001 |
| ≥ 61 | 0.92 (0.74–1.16) | 0.085 | 0.90 (0.70–1.14) | 0.062 |
| Education | | | | |
| Higher | Ref | | Ref | |
| Secondary | 2.38 (1.42–3.53) | <0.001 | 2.32 (1.40–3.33) | <0.001 |
| Primary | 2.62 (1.52–3.74) | <0.001 | 2.67 (1.56–3.80) | <0.001 |
| Illiterate | 2.77 (1.68–3.91) | <0.001 | 2.93 (1.78–4.11) | <0.001 |
| Socioeconomic status | | | | |
| Low | Ref | | Ref | |
| Middle | 1.68 (1.28–2.12) | <0.001 | 1.57 (1.22–1.95) | <0.001 |
| High | 2.89 (2.25–3.54) | <0.001 | 2.63 (2.12–3.14) | <0.001 |
| Physical activity | | | | |
| Adequate/high | Ref | | Ref | |
| Low | 1.52 (1.18–1.87) | <0.001 | 1.82 (1.28–2.34) | <0.001 |

Multinomial logistic regression analysis was done to assess the relationship of covariates with general and abdominal obesity. In the model, general obesity and abdominal obesity (yes) were the dependent variable and socio-demographic variables were the independent variable.

prevalence of general obesity was found to have increased almost 3 fold in women aged 25–34 years, from 1.37% and 7.47% in 1999 to 8.28% and 21.23% in 2014 in rural and urban areas in Bangladesh, respectively [23].

In this study, the prevalence of abdominal obesity was 49% with 38.1% in rural areas and 58.6% in urban areas. In a recent study, the prevalence of abdominal obesity was found as 48.7% in rural Bangladeshi women [6] which is higher than the prevalence found in rural participants but lower than the urban participants in our study. The high prevalence of abdominal obesity (49%) in the present study indicates that a significant portion of adult women may not be classified as obese only based on BMI levels. So, a single BMI cut-off value might not be sufficient to define general obesity in women. Age, sex and ethnic-specific BMI levels should be considered in defining general obesity.

In countries like Bangladesh, an increasing trend of obesity may be attributed to the rapid urbanization, availability of fast and processed food, sedentary or less physically active lifestyle, use of mechanized transport and consumption of energy-rich but a nutrient-poor diet. These determinants are more common in city areas that might be a reason for the higher prevalence of general and abdominal obesity among urban residents.

In a previous study, a high prevalence of obesity was associated with the middle age, female gender, higher economic and educational level and low physical activity in the South Asian region [24]. Similar findings were observed in the present study, except for the education level. In our study, the prevalence of abdominal obesity was higher than in males found in a previous study in Bangladesh [6]. A high prevalence of general and abdominal obesity has also been reported among female subjects in South India [25]. Increased menopause, parity and high intake of the oral contraceptive pill could be the possible contributing factors of increased

prevalence of abdominal obesity in women. The present study findings indicate that our study subjects are at risk of cardio-metabolic diseases. We agree with the statement that an effective intervention strategy with community-based approaches is needed to prevent and treat obesity [6]. In the present study, we observed a higher prevalence of obesity among aged (30 to 50 years) participants, whereas, this prevalence is more common among aged populations in the western nations [26]. In this study, we observed that participants with a higher socioeconomic status and low education levels had an increased percentage of obesity. A similar finding was found in other studies conducted in Canada, Australia and Greece [21,27,28]. Further, large-scale studies are required to know the exact reason for this contrasting finding between socio-economic status and education levels. A possible explanation of this finding might be that the majority of our participants were housewives and formal education was not mandatory for them. Due to increased growth in the economy and development in the agriculture sectors in the last few years, they have a good income in their family. As a result, they can afford food but due to lack of education, they are not aware of nutrition and a healthy diet [6]. They eat rice two to three times per day and consume more sugar and oil with food instead of more fruits and vegetables. All these food items are known to increase the risk of weight gain. Evidence showed that social and dietary factors and physical inactivity promote obesity globally [6]. Physical inactivity, low educational level and high socioeconomic status were significant determinants for general and abdominal obesity in our study. Smoking was found to be a protective factor for general and abdominal obesity [27]. However, in this study, the participants were not used to smoking as Bangladeshi women are generally not familiar with smoking.

The strengths of this study were that we recruited the participants from four administrative regions of Bangladesh in terms of geography, place of residence, socioeconomic status, education levels and physical activity. This study perhaps is the first to estimate the prevalence of both general and abdominal obesity in adult women living in the rural and urban areas of Bangladesh. This study has some limitations. First, the cross-sectional nature of the study does not allow for the cause-effect relationship to be established. Therefore, we cannot demonstrate that identified risk factors are causally related to both general and abdominal obesity. Second, we did not include information on individual dietary habits in detail which may be another determinant of obesity. Third, the sample size was relatively small. Although we invited a large number of participants from the Sylhet region, a significant portion of the participants did not agree to participate in the study. The study has been conducted in four regions of Bangladesh; therefore, we recommend a large-scale study including all divisional regions to explore the exact scenario of obesity prevalence in adult women in Bangladesh.

## Conclusions

This study reports an increased prevalence of general and abdominal obesity in women not only in urban areas but also in rural areas in Bangladesh. Low physical activity, middle age, high socioeconomic status and low education level were associated with the increased prevalence of general and abdominal obesity. Such a high prevalence of general and abdominal obesity is a health concern for Bangladeshi adults. Moreover, both conditions are known to be associated with increased risk of morbidity and mortality including a higher risk of thriving non-communicable diseases. Therefore, building public awareness and effective health intervention strategies are urgently needed to address these conditions.

## Acknowledgments

The authors would like to thank all volunteers for their active cooperation in the study.

## Author Contributions

**Conceptualization:** Farjana Islam.

**Data curation:** Rahanuma Raihanu Kathak, Abu Hasan Sumon.

**Formal analysis:** Abu Hasan Sumon, Noyan Hossain Molla.

**Funding acquisition:** Farjana Islam.

**Investigation:** Farjana Islam, Rahanuma Raihanu Kathak, Noyan Hossain Molla.

**Methodology:** Rahanuma Raihanu Kathak, Abu Hasan Sumon, Noyan Hossain Molla.

**Software:** Abu Hasan Sumon.

**Supervision:** Farjana Islam.

**Validation:** Noyan Hossain Molla.

**Writing – original draft:** Farjana Islam.

**Writing – review & editing:** Farjana Islam.

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
