## [Decision Letter · Decision Letter 0]

13 Mar 2020

PONE-D-19-33594

Prevalence and associated risk factors of general and abdominal obesity in rural and urban women in Bangladesh

PLOS ONE

Dear Mrs. Islam,

Thank you for submitting your manuscript to PLOS ONE. After careful consideration, we feel that it has merit but does not fully meet PLOS ONE’s publication criteria as it currently stands. Therefore, we invite you to submit a revised version of the manuscript that addresses the points raised during the review process.

We would appreciate receiving your revised manuscript by Apr 27 2020 11:59PM. To enhance the reproducibility of your results, we recommend that if applicable you deposit your laboratory protocols in protocols.io, where a protocol can be assigned its own identifier (DOI) such that it can be cited independently in the future. For instructions see: http://journals.plos.org/plosone/s/submission-guidelines#loc-laboratory-protocols

We look forward to receiving your revised manuscript.

Kind regards,

Benn Sartorius, PhD

Academic Editor

PLOS ONE

Additional Editor Comments (if provided):

Please include a completed STROBE checklist in the revised manuscript as a supplementary file.

A checklist relevant to the study design can be found at:

https://www.strobe-statement.org/index.php?id=available-checklists

Journal Requirements:

2. In your Methods section, please provide additional information about the participant recruitment method and the demographic details of your participants. Please ensure you have provided sufficient details to replicate the analyses such as: a) the recruitment date range (month and year), b) a description of any inclusion/exclusion criteria that were applied to participant recruitment, c) a table of relevant demographic details, d) a statement as to whether your sample can be considered representative of a larger population (and a description of the methods used to calculate sample size), e) a description of how participants were recruited, and f) descriptions of where participants were recruited and where the research took place.

3. Our internal editors have looked over your manuscript and determined that it is within the scope of our Determinants, Consequences and Management of Obesity  Call for Papers. This collection of papers is headed by a team of Guest Editors for PLOS ONE:Rachel Nugent and Pratibha V. Nerurkar. Additional information can be found on our announcement page: https://collections.plos.org/s/obesity-one.  

If you would like your manuscript to be considered for this collection, please let us know in your cover letter and we will ensure that your paper is treated as if you were responding to this call. If you would prefer to remove your manuscript from collection consideration, please specify this in the cover letter.

Reviewers' comments:

Reviewer's Responses to Questions

**Comments to the Author**

1. Is the manuscript technically sound, and do the data support the conclusions?

Reviewer #1: Yes

Reviewer #2: Partly

2. Has the statistical analysis been performed appropriately and rigorously? 

Reviewer #1: I Don't Know

Reviewer #2: Yes

3. Have the authors made all data underlying the findings in their manuscript fully available?

Reviewer #1: Yes

Reviewer #2: Yes

4. Is the manuscript presented in an intelligible fashion and written in standard English?

Reviewer #1: Yes

Reviewer #2: No

5. Review Comments to the Author

Reviewer #1: This manuscript provided information on central and overall adiposity among a small group of Bangladeshi women by geographic region. Overall, this paper is somewhat underdeveloped. Further information is needed to justify urban-rural analysis and data presented by 4 regions. Further more information is needed to adequately evaluate the results.

Abstract: “Physical activity, socioeconomic status and education levels were associated with the increased prevalence of general and abdominal obesity.” Need to specify showed increased prevalence, for example physical activity or inactivity, lower or higher SES, lower or higher educ.

Line 16, page 3: “the epidemiological” should be “an epidemiological” not ‘the’

Line 8, page 4: ” The Bangladesh’s population is of this country is projected to increase from 147 million in 2007 to 218 million in 2050. This is super old data using 2007. There is more recent population data.

Methods: Please describe the enrollment process. How were these women selected, etc. Please provide a rationale for the selection of these 4 regions

How was the data found to be able to know medical history (line 1, page 5) since there seemed to be a difference between “self reported or medical history”

Page 6, line 6. Why are you including males in this definition since men are not part of your study.

Important variables need to be defined. Particularly physical activity, smoking status, food habits, and socio-economic,—how was this determined. Only 2 variables were adequately defined: the measurements, educ.

How was urban and rural status determined?

How were the dependent variables defined in the model?

Why is overweight considered important as most of the literature seems to indicate that the concern with overall adiposity is related to obese status and not so much for overweight.

Overall the statistics section was significantly underdeveloped.

Page 6, line 14. “Differences for the baseline data in WC and BMI groups were analyzed by independent sample t-test and ANOVA, respectively.” What is the baseline data? This suggest a longitudinal data collection design? Confusing.

There needs to be some background on the value of assessing both since most of the issues with health conditions are more strongly related to abdominal adiposity. Why both? What does analyzing both add to the literature.

There needs to be some indication of how the model was constructed. It is abit concerning that there are 5 age groups in the model (but table 1 does not show these age groups). As Sylhet has 22 urban people in 5 age groups, then even with equal numbers in each group that would mean there are 4 (or fewer in some groups. This is VERY small numbers to consider reasonable for a finding.

Much of the discussion really focused on obesity yet it seems that the analyses used overweight too as a risky weight since in the results mean BMI was used.

Table 2: the values indicate 2 places with no-one in that region classified as obese. This really begs the question as to whether the overall finding are really a reflection on Bangladesh prevalence. I struggle with understanding the value of 1) urban vs rural and 2) by region, especially given the small number of participants.

Reviewer #2: This article addresses an important public health issue not just in Bangladesh but globally. With many LMIC populations undergoing epidemiological transitions, it is crucial to understand contextual differences in obesity levels in order to tailor interventions appropriately. The comparison between rural and urban is therefore a key addition to the body of knowledge.

The statistical analyses are technically sound and appropriate conclusions have been drawn from the data. However, the overall language and grammar of the manuscript needs a major review for clarity, if possible working with a copy-editor. The methods section in my view also needs further clarification.

MAJOR REVISIONS

1. I recommend that the authors work with a copy-editor in reviewing language and grammar of the entire manuscript. Because the paper not only compares rural and urban but also the different regions, the language needs to be clear and precise to get the right story across. The discussion section will also benefit immensely from language review in order to bring forward the true value of this study.

Examples of unclear areas: page 2 lines 19/20

"An increasing trend of abdominal obesity

19 was found among the participants with increasing age except for the Chattragram region,

20 {whereas}, the prevalence of general obesity was comparatively higher at middle age to 60 years

21 participants."

Also page 7, line 9; page 8 line 6-7.

2. Methods:

a) More detail can be added to the description of the setting eg How were the divisions selected? Out of a total of how many? This helps the reader to see generalisability of the study. What is the population size of these regions? Do they each have the same proportion of rural to urban area?

b) How was the sample size arrived at? The Sylhet region has a noticeably smaller sample size compared to the others - why was this? A discussion of potential implications may be necessary too.

c) Physical activity categories: it might be useful to show the classes of jobs or occupation that were used to arrive at the physical activity groups and whether this follows any other connvetion or tool used in quantifying physical activity. This is especially important because there is a sizeable proportion of housewives - how does this occupation the classification of physical activity.

MINOR REVISIONS

Results: It brings better clarity if the reference category is mentioned when reporting regression results, and to indicate the direction of the association, especially in the abstract.

6. PLOS authors have the option to publish the peer review history of their article (what does this mean?). If published, this will include your full peer review and any attached files.

Reviewer #1: No

Reviewer #2: No

---

## [Author Response · Author response to Decision Letter 0]

4 Apr 2020

Date: 04 April 2020

Prof. Benn Sartorius

Manuscript ID: PONE-D-19-33594

Dear Editor, 

We appreciate the feedback from the editor and reviewers and their helpful suggestions for further improving our manuscript. We have carefully considered all the valuable comments and made revisions accordingly. We have revised the entire manuscript text to avoid grammatical errors. Text changes are marked in colour in the attached revised manuscript. We also formatted the revised manuscript according to the journal style. We feel that we satisfactorily responded to all the issues raised and are looking forward to your reconsideration of our manuscript. 

We thank the referees for evaluating the manuscript and for the helpful suggestions which contributed substantially to improve the quality of our manuscript. In the following, we briefly cite the comments made by the referees and state how we responded to them.

We would be happy to include our manuscript to of special collection “Consequences and Management of Obesity” 

With best regards,

Farjana Islam

Department of Biochemistry and Molecular Biology

Shahjalal University of Science and Technology, Sylhet, Bangladesh

E-mail: farjanaislam308@gmail.com

Response to Additional Editor Comments (if provided):

Please include a completed STROBE checklist in the revised manuscript as a supplementary file. Response: According to the suggestion STROBE checklist has been included in the manuscript as a supplementary file.

Journal Requirements:

 Please ensure that your manuscript meets PLOS ONE's style requirements, including those for file naming. The PLOS ONE style templates can be found at http://www.plosone.org/attachments/PLOSOne_formatting_sample_main_body.pdf and http://www.plosone.org/attachments/PLOSOne_formatting_sample_title_authors_affiliations.pdf

Response: We have formatted the revised manuscript and named the files according to the journal style.

2. In your Methods section, please provide additional information about the participant recruitment method and the demographic details of your participants. Please ensure you have provided sufficient details to replicate the analyses such as: a) the recruitment date range (month and year), b) a description of any inclusion/exclusion criteria that were applied to participant recruitment, c) a table of relevant demographic details, d) a statement as to whether your sample can be considered representative of a larger population (and a description of the methods used to calculate sample size), e) a description of how participants were recruited, and f) descriptions of where participants were recruited and where the research took place.

 Response: According to the suggestion, we have included details in the method section (page 4-5).

3. Our internal editors have looked over your manuscript and determined that it is within the scope of our Determinants, Consequences and Management of Obesity Call for Papers. This collection of papers is headed by a team of Guest Editors for PLOS ONE:Rachel Nugent and Pratibha V. Nerurkar. Additional information can be found on our announcement page: https://collections.plos.org/s/obesity-one. 

If you would like your manuscript to be considered for this collection, please let us know in your cover letter and we will ensure that your paper is treated as if you were responding to this call. If you would prefer to remove your manuscript from collection consideration, please specify this in the cover letter.

Response: Thanks for the suggestion. We would be happy to include our manuscript to of special collection “Consequences and Management of Obesity” 

Response: All authors names and the affiliated institute name have been checked to avoid errors.

List of changes made in response to the reviewers

Responses to Reviewer #1

Reviewer #1: This manuscript provided information on central and overall adiposity among a small group of Bangladeshi women by geographic region. Overall, this paper is somewhat underdeveloped. Further information is needed to justify urban-rural analysis and data presented by 4 regions. Further more information is needed to adequately evaluate the results.

Response: Thanks for the comments. We have considered all comments during the revision and further improvement of the manuscript.

Abstract: “Physical activity, socioeconomic status and education levels were associated with the increased prevalence of general and abdominal obesity.” Need to specify showed increased prevalence, for example physical activity or inactivity, lower or higher SES, lower or higher educ.

Response: Thanks for the suggestion. The sentences in the abstract section have been corrected. Regarding food habits, we asked participants about their daily and weekly habits but a significant number of the participants did not provide detail information on their food habits. Therefore, we could not include individual dietary habits in detail. But we assume that food habits may be another determinant of obesity in Bangladeshi women (page 10, line 2).

Line 16, page 3: “the epidemiological” should be “an epidemiological” not ‘the’

Response: The word has been corrected based on suggestion.

Line 8, page 4: ” The Bangladesh’s population is of this country is projected to increase from 

Response: The sentence has been corrected.

147 million in 2007 to 218 million in 2050. This is super old data using 2007. There is more recent population data.

Response: Thanks for the suggestion. The current population size in Bangladesh is 164 million and projected to increase 192 million by 2050. The information has been updated in the text (page 4, lines 11-12). 

Methods: Please describe the enrollment process. How were these women selected, etc. Please provide a rationale for the selection of these 4 regions

Response: The selected four regions were the main administrative regions in Bangladesh which were located in the central (Dhaka, the capital city), Northwest (Rajshahi), South (Chittagong) and East (Sylhet) of Bangladesh. Details on participants enrolled have been included in the method section (page 5, lines 5-15).

How was the data found to be able to know medical history (line 1, page 5) since there seemed to be a difference between “self reported or medical history”

Response: Thanks for the remark and we agree with that. As exclusion criteria, we excluded the participants who had known cardiac, renal, liver and malignant diseases during data collection. We asked the participants whether they have diagnosed as positive by the physician for any of these diseases. To avoid the confusion we have now edited the sentence (page 5 lines 13-15). 

Page 6, line 6. Why are you including males in this definition since men are not part of your study. 

Response: We included the definition for both genders which would be helpful to understand the differences of the specific variable level between male and female subjects.

Important variables need to be defined. Particularly physical activity, smoking status, food habits, and socio-economic,—how was this determined. Only 2 variables were adequately defined: the measurements, educ.

Response: Response: According to the suggestion, physical activity, smoking status, food habits and socio-economic status have been defined in the method section (page 6 line 23, page 7 lines 1-3).

How was urban and rural status determined?

Response: In our study, the city areas were considered the urban areas where people are mainly involved with a different types of works and jobs. On the other hand, the areas outside/far from the city were considered as rural areas where a major percentage of people are involved with the agricultural works. 

How were the dependent variables defined in the model?

Response: In the regression model, general and abdominal obesity (yes) was considered as the dependent variable and socio-demographic variables as the independent variable (details in the statistics section, page 7).

Why is overweight considered important as most of the literature seems to indicate that the concern with overall adiposity is related to obese status and not so much for overweight.

Response: We agree that obesity is the major concern for public health other than overweight. However, participants with overweight have a future trend to develop obesity in their life. Therefore, besides obesity, it might be important to present the percentage overweight among the participants. However, considering the suggestion, we have edited the text and given the emphasis on obesity other than overweight throughout the manuscript. 

Overall the statistics section was significantly underdeveloped.

Response: We have checked and revised the statistical section to avoid any missing information (page 7). 

Page 6, line 14. “Differences for the baseline data in WC and BMI groups were analyzed by independent sample t-test and ANOVA, respectively.” What is the baseline data? This suggest a longitudinal data collection design? Confusing.

Response: Thanks for marking the errors. It was our mistake in wording the term. It would be socio-demographic characteristics/data instead of baseline data. It has been corrected in the statistics section (page 7).

There needs to be some background on the value of assessing both since most of the issues with health conditions are more strongly related to abdominal adiposity. Why both? What does analyzing both add to the literature.

Response: In some studies, measures of abdominal obesity such as WC indicated as the best parameter correlated with cardiovascular diseases and metabolic dysfunctions. While some other studies have not found sufficient evidence that the measure of abdominal obesity is superior to BMI in predicting the risk of cardiovascular disorders (Tailor et al, Am J Clin Nutr. 2010, 91: 547-556. 10.3945/ajcn.2009.28757; Liu et al. BMC Public Health. 2011, 11: 35-10.1186/1471-2458-11-, Qiao et al. Eur J Clin Nutr. 2010, 64: 30-34. 10.1038/ejcn.2009.93. and some other studies. Moreover, some studies indicated the association of both general and abdominal obesity with cardiovascular diseases and metabolic dysfunctions. Considering the previous reports, we have analyzed both general and abdominal obesity in our participants. Some background has been added in the introduction section (page 4, lines 1-4). 

There needs to be some indication of how the model was constructed. It is a bit concerning that there are 5 age groups in the model (but table 1 does not show these age groups). As Sylhet has 22 urban people in 5 age groups, then even with equal numbers in each group that would mean there are 4 (or fewer in some groups. This is VERY small numbers to consider reasonable for a finding.

Response: In regression analysis, the model was constructed for all participants enrolled from the four regions not for the individual region. According to the age range, we divided participants into five groups to see which age groups are more associated with the prevalence of obesity. In such analysis, we observed that the middle age groups were significantly associated with the prevalence of obesity. We also checked the analysis even excluding the small number of participants (n=22) from the urban area of the Sylhet region, but the results remained significant. 

Much of the discussion really focused on obesity yet it seems that the analyses used overweight too as a risky weight since in the results mean BMI was used.

Response: According to the comments we have revised the text in the discussion section and given the emphasis on obesity other than overweight.

Table 2: the values indicate 2 places with no-one in that region classified as obese. This really begs the question as to whether the overall finding are really a reflection on Bangladesh prevalence. I struggle with understanding the value of 1) urban vs rural and 2) by region, especially given the small number of participants.

Response: Thanks for the comments. As a limitation of the study, we already mentioned in the text that the present study has been conducted in four regions of Bangladesh with a small number of participants; therefore, we recommend a further study including all regions of Bangladesh to explore the exact scenario of obesity prevalence in adult women in Bangladesh (page 12 lines 21-23). In the present study, we could not collect large data due to limited funding. However, in our next study, we planned to collect data on both genders including young adults from all eight regions of Bangladesh to estimate the prevalence of hypertension, diabetes and obesity and their associated risk factors. 

Reviewer #2: This article addresses an important public health issue not just in Bangladesh but globally. With many LMIC populations undergoing epidemiological transitions, it is crucial to understand contextual differences in obesity levels in order to tailor interventions appropriately. The comparison between rural and urban is therefore a key addition to the body of knowledge.

Response: Thank you for appreciating comments about our study.

The statistical analyses are technically sound and appropriate conclusions have been drawn from the data. However, the overall language and grammar of the manuscript needs a major review for clarity, if possible working with a copy-editor. The methods section in my view also needs further clarification.

Response: We appreciate the positive comments on the statistics and conclusion section. We thank you for the valuable suggestions on language and grammar checking of the manuscript. During the revision of the manuscript, we have taken help from a colleague of the English Department and a senior colleague who is experienced in scientific writing from our institute.

MAJOR REVISIONS

1. I recommend that the authors work with a copy-editor in reviewing language and grammar of the entire manuscript. Because the paper not only compares rural and urban but also the different regions, the language needs to be clear and precise to get the right story across. The discussion section will also benefit immensely from language review in order to bring forward the true value of this study.

Response: According to the suggestion, we have taken help from a colleague of the English Department and a senior colleague who is experienced in scientific writing from our institute. With their help we have revised the entire manuscript to avoid language and grammatical errors. 

Examples of unclear areas: page 2 lines 19/20

"An increasing trend of abdominal obesity

19 was found among the participants with increasing age except for the Chattragram region,

20 {whereas}, the prevalence of general obesity was comparatively higher at middle age to 60 years 21 participants."

Response: The sentence has been revised and made simple to clear the meaning (page 2 lines 18-19).

Also page 7, line 9; page 8 line 6-7.

Response: The sentences have been revised (page 8 lines 3-4). In page 8, the sentences have been revised and shortened to clear the meaning (page 8 lines 20-22).

2. Methods:

a) More detail can be added to the description of the setting eg How were the divisions selected? Out of a total of how many? This helps the reader to see generalisability of the study. What is the population size of these regions? Do they each have the same proportion of rural to urban area?

Response: In Bangladesh, there are eight administrative divisional regions. The selected four regions were the larger and major administrative regions in Bangladesh which were located in the central (Dhaka: capital city), Northwest (Rajshahi), South (Chittagong) and East (Sylhet) of Bangladesh (information has been added in the method section, page 5). The population size was about 5 million in Chittagram, 6 million in Dhaka, 1.7 million in Rajshahi and 1.9 million in Sylhet region. Dhaka is the capital city in Bangladesh, so a major portion of this region is an urban area. In other regions, the proportion of rural areas is also high but the difference is not big. Now a day’s, urban areas in Bangladesh are increasing due to rapid urbanization and industrialization. But still now, a major portion (about 60%) of the population lives in rural areas in Bangladesh.

b) How was the sample size arrived at? The Sylhet region has a noticeably smaller sample size compared to the others - why was this? A discussion of potential implications may be necessary too.

Response: Details about the participant’s enrollment process have been included in the method section (page 5). However, we also believe that the sample size is relatively small and large data should be collected from all regions of Bangladesh to explore the exact prevalence throughout the country. Regarding sample size, we have included it as one of the limitations of our study (page 12 line 19-21). We also requested a large number of subjects to participate from the Sylhet region, unfortunately a major portion of them was not agreed to participate and not willing to disclose their personal information Moreover, many of the participants did not provide complete information in the questionnaire form. So we did not include them due to missing information. We have included some text in the discussion section (page 12 line 19-21). 

c) Physical activity categories: it might be useful to show the classes of jobs or occupation that were used to arrive at the physical activity groups and whether this follows any other connvetion or tool used in quantifying physical activity. This is especially important because there is a sizeable proportion of housewives - how does this occupation the classification of physical activity.

Response: Thanks for the suggestion. Now physical activities have been defined with examples (page 6 line 23 and page 7 lines 1-2).

MINOR REVISIONS

Results: It brings better clarity if the reference category is mentioned when reporting regression results, and to indicate the direction of the association, especially in the abstract.

Response: According to suggestion the reference category has been added in the abstract section (page 2 lines 20-22).

---

## [Decision Letter · Decision Letter 1]

4 May 2020

PONE-D-19-33594R1

Prevalence and associated risk factors of general and abdominal obesity in rural and urban women in Bangladesh

PLOS ONE

Dear Mrs. Islam,

Thank you for submitting your manuscript to PLOS ONE. After careful consideration, we feel that it has merit but does not fully meet PLOS ONE’s publication criteria as it currently stands. Therefore, we invite you to submit a revised version of the manuscript that addresses the points raised during the review process.

We would appreciate receiving your revised manuscript by Jun 18 2020 11:59PM. To enhance the reproducibility of your results, we recommend that if applicable you deposit your laboratory protocols in protocols.io, where a protocol can be assigned its own identifier (DOI) such that it can be cited independently in the future. For instructions see: http://journals.plos.org/plosone/s/submission-guidelines#loc-laboratory-protocols

We look forward to receiving your revised manuscript.

Kind regards,

Benn Sartorius, PhD

Academic Editor

PLOS ONE

Reviewers' comments:

Reviewer's Responses to Questions

**Comments to the Author**

1. If the authors have adequately addressed your comments raised in a previous round of review and you feel that this manuscript is now acceptable for publication, you may indicate that here to bypass the “Comments to the Author” section, enter your conflict of interest statement in the “Confidential to Editor” section, and submit your "Accept" recommendation.

Reviewer #2: (No Response)

2. Is the manuscript technically sound, and do the data support the conclusions?

Reviewer #2: Yes

3. Has the statistical analysis been performed appropriately and rigorously? 

Reviewer #2: Yes

4. Have the authors made all data underlying the findings in their manuscript fully available?

Reviewer #2: Yes

5. Is the manuscript presented in an intelligible fashion and written in standard English?

Reviewer #2: No

6. Review Comments to the Author

Reviewer #2: The manuscript is now methodologically sound and sufficient discussion has been given regards study weaknesses and further research. However I feel the grammar and language still need further revision to present these finding s in the clearest possible way. Below are more examples that may need to be revised:

• Page 4 line 1-8; line 16-18

• Page 5 line 6: should be “aged above 18”

Line 9-11

• Page 7 line 14-15

• Page 8 line 12: should be “higher prevalence”

• Page 10 line 4-5: “ documented” not documents…”higher than that reported”…

Line 8: “than that reported”

Line 9-11: “the prevalence of general obesity was found to have increased almost 3 fold…”

• Page 11 lines 1-3

Line 16-17

• Page 12 line: should be “a significant portion of the participants did not agree to participate in the study”

7. PLOS authors have the option to publish the peer review history of their article (what does this mean?). If published, this will include your full peer review and any attached files.

Reviewer #2: No

---

## [Author Response · Author response to Decision Letter 1]

6 May 2020

Date: 06 May 2020

Prof. Benn Sartorius

Manuscript ID: PONE-D-19-33594R1

Dear Editor, 

We acknowledge and appreciate the editor and reviewers suggestions for further improving our manuscript. We have carefully considered all the valuable comments and made revisions accordingly. We have revised the entire manuscript text once again to avoid language and grammatical errors. Text changes are marked in colour in the attached revised manuscript. We feel that we satisfactorily responded to all the issues raised and are looking forward to your reconsideration of our manuscript. 

With best regards,

Farjana Islam

Department of Biochemistry and Molecular Biology

Shahjalal University of Science and Technology, Sylhet, Bangladesh

E-mail: farjanaislam308@gmail.com

List of changes made in response to the reviewers

Responses to Reviewer #2

Reviewer #2: The manuscript is now methodologically sound and sufficient discussion has been given regards study weaknesses and further research. However I feel the grammar and language still need further revision to present these finding s in the clearest possible way. Below are more examples that may need to be revised:

Response: We appreciate the valuable comments and suggestions. According to the suggestion we have revised the manuscript text to avoid the language and grammatical errors.

• Page 4 line 1-8; line 16-18

Response: The text has been revised (page 4 line 1-9 and line 15-16)

• Page 5 line 6: should be “aged above 18”

Line 9-11

Response: The word has been corrected according to suggestion. Moreover, lines 9-11 have been revised to clear the meaning (page 5 line 9-11)

• Page 7 line 14-15

Response: The sentences have been revised (page 7 line 13-14).

• Page 8 line 12: should be “higher prevalence”

Response: Thanks for marking the error. This has now been corrected. 

• Page 10 line 4-5: “ documented” not documents…”higher than that reported”…

Line 8: “than that reported”

Line 9-11: “the prevalence of general obesity was found to have increased almost 3 fold…”

Response: The words have been corrected according to the suggestion (Page 10 line 3-9)

• Page 11 lines 1-3

Line 16-17

Response: The sentences have been revised. (page 11 line 1-2 and line 13-16).

• Page 12 line: should be “a significant portion of the participants did not agree to participate in the study”

Response: The sentence has been edited based on suggestion.

---

## [Editor Report · Decision Letter 2]

13 May 2020

Prevalence and associated risk factors of general and abdominal obesity in rural and urban women in Bangladesh

PONE-D-19-33594R2

Dear Dr. Islam,

We are pleased to inform you that your manuscript has been judged scientifically suitable for publication and will be formally accepted for publication once it complies with all outstanding technical requirements.

With kind regards,

Benn Sartorius, PhD

Academic Editor

PLOS ONE
---

## [Editor Report · Acceptance letter]

20 May 2020

PONE-D-19-33594R2 

Prevalence and associated risk factors of general and abdominal obesity in rural and urban women in Bangladesh 

Dear Dr. Islam:

I am pleased to inform you that your manuscript has been deemed suitable for publication in PLOS ONE. Congratulations! Your manuscript is now with our production department. 

With kind regards,

on behalf of

Dr. Benn Sartorius 

Academic Editor

PLOS ONE